# Unequal access to opioid agonist treatment and sterile injecting equipment among hospitalized patients with injection drug use-associated infective endocarditis

**Thomas D. Brothers**[1,2]*, **Kimiko Mosseler**[3], **Susan Kirkland**[1,4], **Patti Melanson**[5†], **Lisa Barrett**[1,6], **Duncan Webster**[1,7]

**1** Department of Medicine, Dalhousie University, Halifax, Nova Scotia, Canada, **2** UCL Collaborative Centre for Inclusion Heath, Institute of Epidemiology and Health Care, University College London, London, United Kingdom, **3** Dalhousie Medicine New Brunswick, Dalhousie University, Saint John, New Brunswick, Canada, **4** Department of Community Health & Epidemiology, Dalhousie University, Halifax, Nova Scotia, Canada, **5** Mobile Outreach Street Health (MOSH), Halifax, Nova Scotia, Canada, **6** Division of Infectious Diseases, Nova Scotia Health, Halifax, Nova Scotia, Canada, **7** Division of Infectious Diseases, Saint John Regional Hospital and Dalhousie University, Saint John, New Brunswick, Canada

† Deceased.
* thomas.brothers.20@ucl.ac.uk

**Data Availability Statement:** All relevant data are within the paper and its Supporting Information files.

## Abstract

### Background

Addiction treatment and harm reduction services reduce risks of death and re-infection among patients with injection drug use-associated infective endocarditis (IDU-IE), but these are not offered at many hospitals. Among hospitalized patients with IDU-IE at the two tertiary-care hospitals in the Canadian Maritimes, we aimed to identify (1) the availability of opioid agonist treatment (OAT) and sterile drug injecting equipment, and (2) indicators of potential unmet addiction care needs.

### Methods

Retrospective review of IDU-IE hospitalizations at Queen Elizabeth II Health Sciences Centre (Halifax, Nova Scotia) and the Saint John Regional Hospital (Saint John, New Brunswick), October 2015 -March 2017. In Halifax, there are no addiction medicine providers on staff; in Saint John, infectious diseases physicians also practice addiction medicine. Inclusion criteria were: (1) probable or definite IE as defined by the modified Duke criteria; and (2) injection drug use within the prior 3 months.

### Results

We identified 38 hospitalizations (21 in Halifax and 17 in Saint John), for 30 unique patients. Among patients with IDU-IE and untreated opioid use disorder, OAT was offered to 36% (5/14) of patients in Halifax and 100% (6/6) of patients in Saint John. Once it was offered, most patients at both sites initiated OAT and planned to continue it after discharge. In Halifax, no patients were offered sterile injecting equipment, and during five hospitalizations staff

**Funding:** This work was supported by the Ross Stewart Smith Memorial Fellowship in Medical Research from Dalhousie University Faculty of Medicine and the Hui Lee Health Promotion Scholarship from the Canadian Society of Internal Medicine, to TDB. TDB is currently supported by the Dalhousie University Internal Medicine Research Foundation Fellowship, a Canadian Institutes of Health Research Fellowship (CIHR-FRN# 171259), and through the Research in Addiction Medicine Scholars (RAMS) Program (National Institute of Health/National Institute on Drug Abuse; R25DA033211). KM is supported by the Dalhousie Medical Research Foundation Katelyn Robarts Studentship. The funders had no role in study design, data collection and analysis, decision to publish, or preparation of the manuscript.

**Competing interests:** The authors have declared that no competing interests exist.

confiscated patients' own equipment. In Saint John, four patients were offered (and one was provided) injecting equipment in hospital, and during two hospitalizations staff confiscated patients' own equipment. Concerns regarding undertreated pain or opioid withdrawal were documented during 66% (25/38) of hospitalizations, and in-hospital illicit or non-medical drug use during 32% (12/38). Two patients at each site (11%; 4/38) had self-directed discharges against medical advice.

## Conclusions

Patients with IDU-IE in the Canadian Maritimes have unequal access to evidence-based addiction care depending on where they are hospitalized, which differs from the community-based standard of care. Indicators of potential unmet addiction care needs in hospital were common.

## Introduction

North America's complex opioid-related public health crisis is associated with increasing rates of injection drug use [1], and a rapidly rising incidence of injecting-related bacterial and fungal infections, including injection drug use-associated infective endocarditis (IDU-IE) [2–6]. Hospitalization for IDU-IE is a "reachable moment" to effectively engage patients in addiction treatment and harm reduction care, including opioid agonist treatment (OAT; e.g. methadone, buprenorphine) and access to sterile drug injecting equipment [7–11]. OAT is associated with large reductions in all-cause mortality among patients with IDU-IE [12] and may be associated with reduced risk of IDU-IE recurrence [13–16]. Use of sterile injecting equipment is associated with decreased risk of IDU-IE [17]. Although these practices comprise the standard-of-care for opioid use disorder [7, 18], OAT and sterile injecting equipment are not routinely offered in many hospitals; this represents missed opportunities to reduce risks of death and recurrent infections [8, 11, 19–24].

Hospitals differ in the care they offer patients with medical complications of addiction and injection drug use, including IDU-IE [7, 25–27]. In the Canadian Maritime provinces, two academic, tertiary-care medical centres (one in Halifax, Nova Scotia, and one in Saint John, New Brunswick) provide specialist care to patients with infective endocarditis (IE). Both hospitals offer consultation services in infectious diseases, cardiology, and cardiac surgery. Neither hospital offers a specialist addiction medicine consultation service, but the infectious diseases service in Saint John initiates OAT for patients with IDU-IE and assisted in establishing a policy to enable distribution of sterile injecting equipment in hospital. Access to addiction care and harm reduction services has not been evaluated at either hospital.

We aimed to describe the availability and uptake of two evidence-based addiction care practices–OAT and sterile injecting equipment distribution–among patients hospitalized with IDU-IE in Halifax and in Saint John. As a secondary, exploratory objective, we aimed to identify descriptions of unmet care needs documented in the medical records of patients with IDU-IE at each hospital.

## Materials and methods

### Setting and data sources

We performed a retrospective review of hospital inpatients with IDU-IE, admitted from October 1, 2015, to March 31, 2017 (18 months), at the two large, academic, tertiary-care hospitals

in Halifax, Nova Scotia, and Saint John, New Brunswick, that provide definitive care to patients with IE in the Canadian Maritime provinces. In both hospitals, patients with endocarditis may initially be admitted to internal medicine or to cardiology wards, and these patients might be transferred between the two services or to family medicine wards during a prolonged hospital stay.

One investigator at each site (TDB and KM) performed structured chart review, extracting information on demographic and clinical factors, substance use, and treatment history. This information was extracted from multidisciplinary team progress notes, nursing communication notes, specialist consultation records, physician orders, medication administration records, and hospital discharge summaries. All documentation in Halifax was available as paper charts scanned to an online system, and in Saint John there was a combination of physical paper charts and electronic progress and nursing communication notes.

In Halifax, Nova Scotia, the Queen Elizabeth II Health Sciences Centre is a 661-bed university-affiliated tertiary-care centre that serves as the referral centre for patients with IE across Nova Scotia. The hospital has no addiction medicine expertise readily available in the hospital, which is typical of many North American hospitals. OAT is provided through ad hoc calls to community-based addiction physicians or patients' family physicians outside the hospital. No local clinical guidelines or hospital policies exist to advise on caring for people who inject drugs when admitted to hospital.

In Saint John, New Brunswick, the Saint John Regional Hospital is a 524-bed university-affiliated tertiary-care hospital that serves as the referral centre for patients with IE in New Brunswick. Since the early 2000s, it has developed a growing culture towards implementing harm reduction strategies in the hospital inpatient setting. Since 2008, a clinical initiative led by the Division of Infectious Diseases has focused on identifying patients with opioid use disorder in hospital, initiating OAT in hospital and transitioning patients to outpatient addiction treatment services upon discharge [28]. An inpatient program to distribute sterile injecting equipment on the internal medicine ward was established in 2007. While this policy has never been formally evaluated, based on anecdotal reports we expected it to be applied reliably for internal medicine inpatients and applied inconsistently on other wards in the hospital. The version of the relevant policy ("Intravenous Needle Exchange Policy") that was active during the study period is presented in Fig 1. Health professionals (primarily nurses) who offered sterile injecting equipment to inpatients in Saint John were expected to document this in either multidisciplinary progress notes or in electronic nursing communication notes.

In both provinces, costs of OAT are covered by public medication insurance programs. These insurance programs are available free-of-charge to people on social assistance or income assistance; otherwise, there is a deductible proportional to income. During the study period, both provinces' public medication insurance programs requested that patients trial methadone before they would pay for buprenorphine; this has since been changed as both medications are considered first line OAT options. Across Canada, until 2017 physicians required a special certification from Health Canada to initiate methadone for OAT; in Halifax, no hospital-based physicians held this certification, while in Saint John, the infectious diseases physicians and a small number of other specialists, hospitalists, and psychiatrists also held this certification [29]. In both Halifax and Saint John, outpatient OAT is available immediately after hospital discharge for patients with IDU-IE, without a waiting list. Similar to the rest of Canada, OAT is typically prescribed in primary care or in specialized clinics, with dispensing at community pharmacies [29]. In more rural areas of Nova Scotia or New Brunswick, outpatient OAT may not be available in local communities. Both Nova Scotia and New Brunswick have provincially funded programs to distribute sterile drug injecting equipment through community-based non-governmental organizations.

**INTRAVENOUS NEEDLE EXCHANGE POLICY, 4C North**

**Staff Protection Initiative**

1. Sharps containers will be provided to all patients that might be using non-prescription parenteral drugs on 4CN.

2. Needles will be provided to patients using parenteral on an as-needed basis, to include periods of time whereupon access might be more difficult and in consideration of patients' frequency of substance abuse.

3. Safe injection sites will not be provided. Nevertheless, privacy will be respected in consideration of the similar privacy afforded other inpatients.

4. In unusual circumstances, the Internal Medicine Nurse Manager, will be consulted for assistance and/or recommendations. An attending physician or departmental chief may be consulted at the manager's discretion.

5. Patients with parenteral substance abuse problems that continue to inject non-prescription drugs will be encouraged to do so any place that is not their bed. This will minimize risk to staff and personnel.

**Fig 1. Policy enabling inpatient needle and syringe distribution at Saint John Regional Hospital during the study period.**

### Participants

Clinical informatics analysts searched administrative records at both institutions to identify hospitalizations for patients with a discharge diagnosis code potentially consistent with IE, via International Classification of Diseases, Tenth Revision (ICD-10) codes (S1 Table in S1 File). Included cases met both inclusion criteria: (1) probable or definite infective endocarditis as defined by modified Duke criteria [30]; and (2) active injection drug use, defined as injection drug use within 3 months of the IE hospitalization documented in the medical record [19]. We chose this definition as it has been used in several studies to identify IDU-IE [4, 19]. Hospitalization episodes that involved patient transfer from the tertiary care hospital to a community hospital and back again (without outpatient discharge) were counted as a single hospitalization.

### Access to opioid agonist treatment (OAT) and sterile injecting equipment

We assessed medical records from IDU-IE hospitalizations to determine the frequency of patients with untreated opioid use disorder, which we defined as documented injection opioid use without an active prescription for OAT (i.e., methadone, buprenorphine-naloxone, or slow-release oral morphine) at the time of patients' presentation to hospital. At both hospitals, patients' self-report of OAT prescription and dosage is confirmed with their community pharmacy before doses are provided. We assessed how often these patients with IDU-IE and

untreated opioid use disorder were (a) offered initiation of OAT (i.e., methadone, buprenorphine-naloxone, or 24-hour slow-release morphine) in hospital; (b) successfully started OAT in hospital; and (c) were provided with a prescription to continue OAT on discharge from hospital. These categories are consistent with prior work on the "cascade of care" for opioid use disorder among hospitalized patients [27, 31].

Among all hospitalizations with IDU-IE, we assessed patient records for documentation of hospital-based healthcare providers offering or providing sterile drug injecting equipment.

Patients who died in hospital were excluded from the analysis of access to OAT and sterile injecting equipment, because many remained critically ill and/or sedated throughout their hospital stay and so would have been unlikely to access either intervention. Our research team considered several other potential indicators of access to addiction care and harm reduction services in hospital (e.g., counseling regarding safer injecting practices and overdose risk reduction), but ultimately, we did not include them as we determined they would not necessarily be reliably documented in patients' medical records.

### Indicators of potential unmet need

For our secondary, exploratory objective we reviewed medical charts for potential indicators of unmet care need, including (a) undertreated pain or opioid withdrawal, (b) illicit/non-medical drug use in hospital, and (c) patient-initiated discharges against medical advice before completion of IDU-IE inpatient treatment.

### Statistical analysis

Descriptive statistics were summarized using Microsoft Excel.

### Ethics approval

Research Ethics Board (REB) requirements were waived by the Nova Scotia Health Authority REB, who determined this project to be quality assessment. This study was approved by the New Brunswick Horizon Health Authority REB.

## Results

A total of 38 hospitalizations for IDU-IE were identified from October 2015 to March 2017, including 21 in Halifax and 17 in Saint John. These hospitalizations comprised 16 unique patients in Halifax and 14 unique patients in Saint John. Demographic, health, and substance use characteristics of the patients at the time of their first episode of IDU-IE during the study period are summarized in Table 1. Patients were generally young at both sites and were more often men in Saint John. All patients in Halifax and most patients in Saint John injected opioids (primarily hydromorphone), and many patients also injected stimulants. One patient died in hospital in Halifax and two patients died in hospital in Saint John.

A further three patients in Halifax and six patients in Saint John were hospitalized with IE during the study period with a documented remote history (>3 months prior) of injection drug use, so were excluded from this study. At the Halifax site only, data was captured on the total number of IE hospitalizations: the 21 IDU-IE hospitalizations represent 30% of all 71 IE hospitalizations during the study period in Halifax.

We identified the subgroup of hospitalizations for patients with IDU-IE and untreated opioid use disorder, who would have an indication to initiate OAT. In Halifax, there were 19 hospitalizations for patients with IDU-IE and documented opioid injection use who survived to hospital discharge; five (26%) of these patients were already on OAT at the time of hospital

**Table 1. Descriptive characteristics of patients during first episode of injection drug use-associated infective endocarditis in Halifax and Saint John, October 2015-March 2017.**

| | Halifax | Saint John |
|---|---|---|
| Patients | 16 | 14 |
| Average no. of admissions per patient during study period[1] | 1.3 | 1.1 |
| Age, mean (SD) | 32 ± 9 | 36 ± 12 |
| Women | 8 (50%) | 5 (36%) |
| History of infective endocarditis | 7 (44%) | 3 (21%) |
| HIV seropositive | 1 (6%) | 0 (0%) |
| Hepatitis C virus seropositive | 10 (63%) | 10 (71%) |
| Experiencing homelessness or unstable housing[2] | 4 (25%) | 0 (0%) |
| Injection opioid use documented | 15 (100%) [N = 15 with substance use documentation][3] | 12 (92%) [N = 13 with substance use documentation][3] |
| Distribution of opioid use by type | | |
| Hydromorphone[3] | 12 (80%) | 10 (83%) |
| Oxycodone[3] | 0 (0%) | 4 (33%) |
| Morphine[3] | 1 (7%) | 0 (0%) |
| Heroin | 0 (0%) | 0 (0%) |
| Opioid, type not specified | 3 (20%) | 2 (14%) |
| Injection stimulant use documented | 7 (47%) [N = 15 with substance use documentation][3] | 9 (64%) [N = 13 with substance use documentation][3] |
| Distribution of stimulant use by type | | |
| Cocaine | 6 (40%) | 9 (100%) |
| Methamphetamine | 2 (13%) | 5 (55%) |
| Prescription stimulants[4] (e.g. methylphenidate) | 3 (23%) | 0 (0%) |
| Receiving OAT at time of admission | 6 (40%) [N = 15 with documented opioid use] | 8 (67%) [N = 12 with documented opioid use] |
| Distribution of OAT by type | | |
| Methadone | 5 (83%) | 8 (100%) |
| Buprenorphine-naloxone | 1 (17%) | 0 (0%) |

[1]Total number of injection drug use-associated infective endocarditis admissions during the study period were 21 in Halifax and 17 in Saint John

[2]As documented by medical team, with no standard definition

[3]One patient from each site had documented recent injection drug use but no specific information on substances used

[4]Including immediate-release and sustained-release formulations, as distinction not clearly documented in medical record

OAT: Opioid agonist treatment (i.e., methadone, buprenorphine-naloxone, or once-daily slow-release morphine).

admission, resulting in 14 eligible hospitalization episodes for patients with IDU-IE and untreated opioid use disorder. In Saint John, there were 16 hospitalizations for patients with IDU-IE and documented opioid injection use who survived to hospital discharge; ten (63%) of these patients were already on OAT at the time of admission, resulting in six eligible hospitalization episodes for patients with IDU-IE and untreated opioid use disorder.

Access and uptake of OAT among these patients with IDU-IE and untreated opioid use disorder is summarized in Fig 2. Five (36%) of 14 patients in Halifax were offered to initiate OAT in hospital, while all six (100%) of six patients in Saint John were offered OAT initiation in hospital. Notably, for two other patients in Halifax, the infectious diseases consultation service recommended OAT but this was not acted upon. One additional patient in Halifax specifically requested to start OAT in hospital; the request was documented but OAT was not offered or provided.

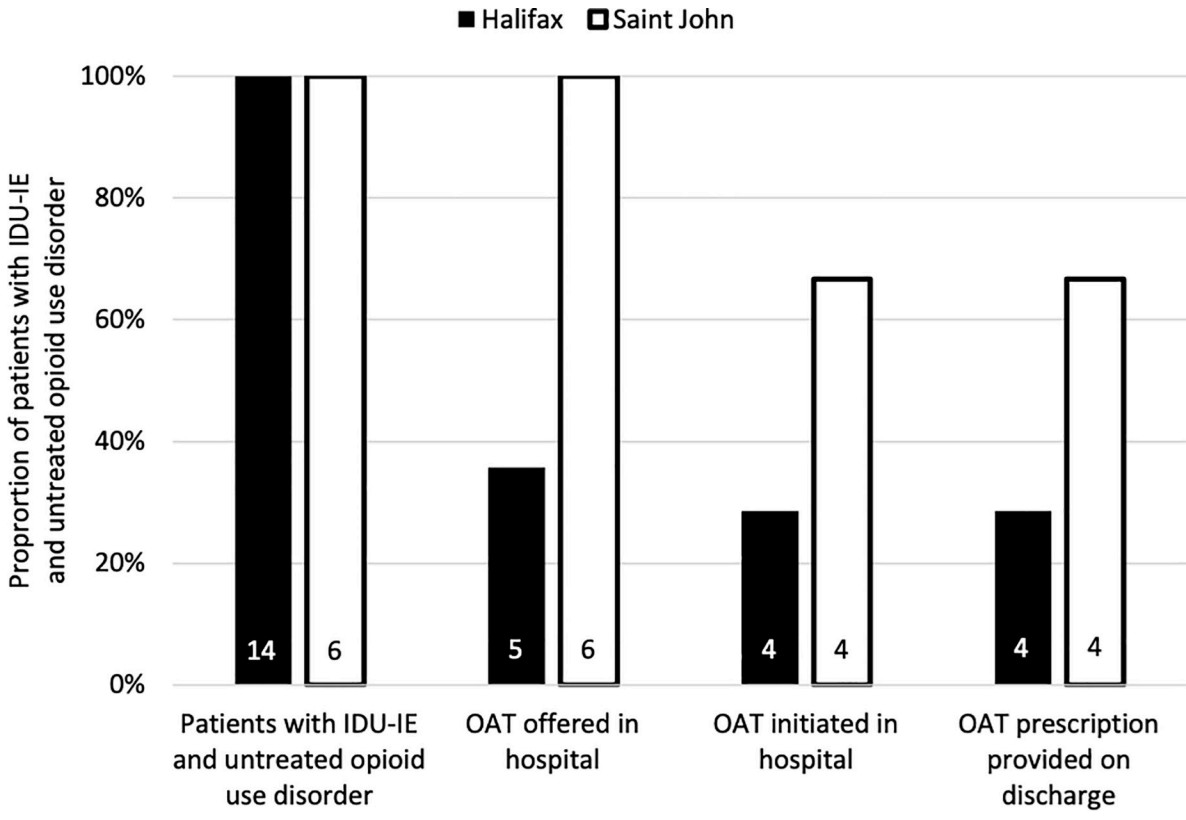

**Fig 2. Access to and uptake of opioid agonist treatment during hospitalizations for injection drug use-associated infective endocarditis in Halifax, Nova Scotia (n = 14 hospitalizations), and Saint John, New Brunswick (n = six hospitalizations), October 2015-March 2017.** Vertical bars represent proportion of patients with IDU-IE and untreated opioid use disorder at each site; numbers within bars represent the number of patients in each category at each site. IDU-IE: Injection drug use-associated infective endocarditis. OAT: Opioid agonist treatment.

In Halifax, no patients had a documented offer or provision of sterile injecting equipment by hospital-based healthcare providers, but during five hospitalizations drug injecting equipment was found in patients' rooms and confiscated by staff. In Saint John, there was documentation during four hospitalizations (27%, out of 15 hospitalizations where the patient survived) that the patient was educated about the inpatient needle and syringe distribution policy, and during one of these hospitalizations (7%) that the patient was provided sterile injecting equipment while in hospital. There were two documented incidents of injecting equipment confiscation by staff in Saint John. Of Saint John patients with IDU-IE admitted to the internal medicine ward (where the inpatient needle and syringe distribution policy was first developed), one out of four patients was offered injecting equipment and no patients had equipment confiscated. The other three Saint John patients were offered equipment while admitted to cardiology or family medicine wards. Within the same cardiology ward, one patient was offered sterile injecting equipment and a different patient had equipment confiscated.

Patients in both hospitals had documentation of other potential unmet care needs related to substance use, summarized in Table 2. Most hospitalizations (66%, 25/38) for IDU-IE at both sites had documentation of patient or staff concerns about patients experiencing uncontrolled pain or undertreated opioid withdrawal. Illicit or non-medical substance use was documented in 31% (12/38) of all IDU-IE hospitalizations, and patient-initiated discharges against medical advice occurred in 11% (4/38) of all IDU-IE hospitalizations.

**Table 2. Frequency of documentation of potential unmet care needs related to substance use and addiction among patients hospitalized with injection drug use-associated infective endocarditis in Halifax, Nova Scotia and Saint John, New Brunswick, Canada.**

|  | Halifax (N = 21 hospitalizations) | Saint John (N = 17 hospitalizations) |
| --- | --- | --- |
| Uncontrolled pain or undertreated opioid withdrawal | 16 (76%) | 9 (53%) |
| Illicit or non-medical substance use in hospital | 7 (33%) | 5 (29%) |
| Patient-initiated discharges against medical advice | 2 (10%) | 2 (12%) |

## Discussion

In this retrospective review of patients with IDU-IE admitted to two tertiary-care hospitals in Canada, we found that patients had differential access to addiction care and that indicators of unmet need were common. Less than half of patients with IDU-IE and untreated opioid use disorder were offered OAT at the hospital in Halifax, Nova Scotia, while all patients were offered OAT at the hospital in Saint John, New Brunswick. We attribute this unequal access to differences in service organization and delivery between the two hospitals: in Halifax, there were no addiction medicine providers on hospital staff, while in Saint John, the infectious diseases consultation service assessed patients with IDU-IE for substance use disorders and initiated OAT. Once it was offered, most patients at both hospitals accepted and initiated OAT. We also found that four patients were offered sterile drug injecting equipment (and one was provided equipment) at the hospital in Saint John, while no patients were offered this in Halifax. Patients at both hospitals had their own injecting equipment confiscated, despite a policy enabling needle and syringe distribution in Saint John.

For patients with IDU-IE, unsafe injection drug use is the underlying cause of their infection and addiction treatment should be incorporated into treatment plans for secondary prevention [5, 32]. All patients with IDU-IE in Halifax and 92% of patients with IDU-IE in Saint John reported opioid injection [33]. Treatment with OAT is the standard-of-care for opioid use disorder; it is associated with large reductions in risk of death [12] and may be associated with decreased risk of readmission [14] among patients with IDU-IE. Accumulating evidence suggests that hospitalization is a "reachable moment" to engage patients in addiction treatment, and that most hospitalized patients with untreated opioid use disorder are interested in initiating OAT [25, 27, 34, 35]. A randomized controlled trial [36] and several cohort studies [15, 16] show that in-hospital initiation of OAT is associated with improved engagement in treatment after hospital discharge, compared to outpatient referrals. Hospitalization with medical complications represents a particularly vulnerable time for people who use illicit or non-prescribed opioids, as the days following hospital discharge are associated with a 4-fold increase in risk of opioid overdose death [37].

Unfortunately, despite these standards of care and convincing evidence of their effectiveness, hospital care for patients with IDU-IE often focuses on management of the infection and its sequelae without addressing the underlying substance use disorder [19, 23, 32, 36, 38, 39]. Our findings of poor access to OAT at the hospital in Halifax are consistent with several other studies from North American hospitals [9]. Among 37 patients admitted to a Maine hospital for IDU-IE who were not on OAT, five (19%) initiated OAT in hospital [40]. Among 28 patients with untreated opioid use disorder undergoing cardiac surgery for IDU-IE in Connecticut, 7 (25%) were offered OAT or naltrexone (an opioid antagonist used for opioid use disorder treatment in the United States) [41]. In a study of 202 patients in Ontario with a first episode of IDU-IE, only 34 (17%) had a prescription for OAT at discharge (but that study did not describe how many patients were already on OAT when admitted) [21]. Among 102 patients admitted to a Boston hospital for IDU-IE, 8% had a plan for medication treatment for

opioid use disorder (including OAT or injectable naltrexone) at the time of discharge [19]. In a U.S. cohort of 1407 patients with untreated opioid use disorder and IDU-IE or injection drug use-associated osteomyelitis, only 44 (3%) were provided with a prescription to continue OAT on discharge [20]. There is limited published quantitative data outside of North America, but qualitative studies from the United Kingdom describe delayed or no access to OAT among hospitalized patients, even for those with an established prescription [24]. In our study, 36% of eligible patients in Halifax and 100% of eligible patients in Saint John were offered to initiate OAT in hospital, and all patients who initiated OAT were provided prescriptions to continue after discharge with community-based outpatient providers.

Several strategies have been described to increase access to addiction care among hospitalized patients [7]. In our study, at the Saint John site every patient with IDU-IE and untreated opioid use disorder was offered OAT because the infectious diseases consultation service considers this within their scope of practice. This model has been successful at other hospitals [42, 43], but addiction care may or may not be available to patients admitted with non-infectious indications. Despite several years passing since our study period, the removal of Health Canada's certification requirement to prescribe methadone for OAT, and Canada's escalating overdose death crisis, the QEII Health Sciences Centre in Halifax still does not have addiction medicine-trained physicians or OAT providers on staff. This is typical of many hospitals in North America. Specialized addiction medicine consultation services can help close this gap, by providing diagnostic and treatment expertise, increasing uptake of these interventions, and assisting in linking hospitalized patients to outpatient addiction treatment [25, 26, 34, 44–46]. In response to identified gaps in care (informed by the present study), hospital-based resident physicians at the Halifax site partnered with community-based addiction physicians to provide an informal (unfunded) addiction medicine consultation service [27, 31]. Clinicians have also begun to call for incorporation of addiction medicine specialists into multidisciplinary endocarditis care teams [8, 47, 48]. Improving access to OAT in general medical settings requires changes in several areas, including improved education for health professionals, prioritizing access to medications on-demand, enabling multiple medication treatment options, and decreasing patients' out-of-pocket costs [31, 49]. Many patients at both hospital sites also reported injecting stimulants; managing stimulant use disorder requires other addiction treatment and harm reduction strategies beyond OAT, including psychosocial interventions and access to sterile injecting equipment [7, 11], which could also be facilitated by specialized inpatient addiction medicine consultation services.

Patients also had differential access to sterile injecting equipment in hospital, and the equipment distribution policy in Saint John was inconsistently implemented. We identified no documented evidence that patients with IDU-IE were offered or provided sterile injecting equipment by hospital-based health care providers in Halifax. In Saint John, where there has been a hospital policy facilitating needle and syringe distribution to hospital inpatients since 2007, patients were offered sterile injecting equipment during four hospitalizations. We did not capture information on reasons why not all patients were offered equipment, and patients may not be made aware of the policy or may feel unsafe disclosing ongoing substance use [39, 50, 51]. Some patients may have stated goals to abstain from injecting, especially as most patients were either already on OAT or initiated OAT in hospital. Many of the patients with IDU-IE were acutely ill during much of their hospital stay, and so the health care team may not have considered the possibility of ongoing substance use in hospital or the potential benefits of offering sterile injecting equipment. Overall, the inconsistent implementation suggests that a written policy is not enough to ensure reliable implementation, which has also been observed at one other Canadian hospital with an inpatient policy to distribute sterile injecting equipment [52].

Several patients at both hospitals were found to have injecting equipment in their hospital rooms and these were confiscated, despite the policy enabling needle and syringe distribution in Saint John and that no hospital policy exists banning sterile injecting equipment in hospital in Halifax. One ward in Saint John offered sterile injecting equipment to one patient with IDU-IE and confiscated it from another. Further, some of this confiscated equipment likely came from the publicly funded needle and syringe distribution programs in both cities; Halifax's program is located just a few blocks away from the hospital. Confiscating needles and syringes leads patients to hide and re-use blunted or contaminated injecting equipment, which increases the risk of bacterial infections including IDU-IE among hospitalized patients who inject drugs [5] and may also increase risk for needle-stick injuries among hospital staff [11, 53]. Coupled with stigmatizing behaviours such as suspicion, surveillance, and restriction of pain medications, these approaches not only erode therapeutic relationships but are ineffective and can cause patients harm [22, 50, 54–58].

Such a setting of mutual mistrust and undertreated pain and opioid withdrawal contributes to high rates of illicit or non-medical substance use in hospital and discharges against medical advice among hospitalized patients who use drugs [22, 54, 55, 58–60]. Hospitals have been conceptualized as high-risk environments for people who use drugs [5, 59], as abstinence-based policies and threats of punishment lead to high-risk behaviours like using drugs alone in locked bathrooms, rushing injections and missing veins, and taking a bigger dose than usual to try to make it last longer [58–60]. We found that documentation regarding illicit or non-medical substance use in hospital was common at both hospital sites, despite universal access to OAT in Saint John. This suggests that OAT is necessary but not sufficient to support the safety of hospitalized patients who inject drugs; policies that support pain management and harm reduction for ongoing substance use in hospital are also needed [7, 11, 50, 51, 53]. The findings of the present study are now informing the development of comprehensive substance use and harm reduction policies at both hospitals. The needle exchange policy in Saint John has since been revised to reframe its focus from a "staff protection initiative" to a component of patient-centered care for people who use drugs. Best practices for hospital inpatient harm reduction policies include the distribution of alcohol swabs, tourniquets, filters, cookers, sterile water, and vitamin C (which may not otherwise be on hospital formularies), in addition to needles and syringes [7, 11]. Hospitalized patients who use drugs should also be provided take-home naloxone kits [7].

Among patients with IDU-IE in both Halifax and Saint John sites who reported injection opioid use, all patients used diverted or non-prescribed pharmaceutical opioids (primarily hydromorphone tablets) and none reported heroin or fentanyl use. This pattern is consistent with the drug supply available in these communities during the study period; nearly all illicit or non-prescribed opioids were pharmaceutical and there has been very little availability of heroin in the Canadian Maritimes for several decades [33, 61, 62]. Illicitly-manufactured fentanyl and its analogues have become more available since the study period, but pharmaceutical opioids still comprise the majority of illicit or non-prescribed opioids in these communities [63]. As all patients in our study had IDU-IE, we cannot say from these data whether injecting pharmaceutical opioid tablets increases risk for serious bacterial infections more than injecting heroin or fentanyl. While one study identified an association between controlled-release hydromorphone prescriptions and IDU-IE in Ontario [64], a small minority of patients with IDU-IE in Ontario have recent prescriptions for hydromorphone [6, 64]. Increasing rates of serious injection drug use-associated bacterial infections have been observed in many jurisdictions [4, 5], and in Ontario appear to parallel transitions in the drug supply towards increasing use of illicitly-manufactured fentanyl [6]. Known risk factors for serious injecting-related infections include more frequent injecting (which has been associated with fentanyl and with

stimulants), re-use of contaminated needles and filters, and subcutaneous injecting [65]. Social determinants, including homelessness (impacting sterile drug preparation) and criminalization and policing practices (leading to rushed, subcutaneous injections), also influence risk [5]. We did not extract information on patients' current or recent outpatient medication prescriptions to determine whether patients had recent prescriptions for these opioids, and we did not have reliable information on housing status. Purposefully prescribing pharmaceutical opioids for injection as an alternative to the unregulated street drug supply, known as "safe supply" prescribing [66, 67], was not known to be a practice in these communities during the study period [62].

## Limitations

Our study has some important limitations. First, our data came from retrospective chart reviews, and it is possible that hospital-based providers offered sterile injecting equipment and did not document this in the medical record. However, in Saint John the needle exchange policy at the time (Fig 1) was framed as a "staff protection initiative" and so providers were expected to document provision of needles to patients. At either site, if sterile injecting equipment were provided surreptitiously and not document, this would still represent a failure to incorporate evidence-based best practices into the care plan. Second, we used self-reported injection opioid use as our operational definition of opioid use disorder (and therefore to identify patients eligible for OAT) since many patients did not have other documentation of diagnostic criteria for substance use disorders. Third, we were only able to identify documentation from health care providers, and so our study does not include the perspectives of patients themselves, and we may have missed patients who did not feel comfortable disclosing their substance use. Fourth, we had a small sample size and a relatively narrow focus on patients with endocarditis, while all hospitalized patients who inject drugs could benefit from the interventions described. We selected this patient population as they represent a group at particularly high risk of death and disability from ongoing unsafe injection drug use, and should be a priority population for facilitating access to evidence-based substance use care. Fifth, our study only included patients at tertiary care centres, and therefore does not necessarily reflect patients or care delivered at community hospitals.

## Conclusion

Patients with IDU-IE in the Canadian Maritimes have unequal access to addiction care depending on where they are hospitalized, which differs from the community-based standard of care. Indicators of potential unmet addiction care needs in hospital are common, including ongoing substance use in hospital. To meet the needs of these patients, hospitals should employ providers with addiction medicine expertise and should develop harm reduction-oriented policies to promote patient safety. Hospital-based addiction care could be improved through integrating addiction medicine and infectious diseases specialist practice (as at the Saint John Regional Hospital), and/or by establishing specialized addiction medicine consultation services and incorporating these providers into multidisciplinary endocarditis care teams. Further research is needed on the most effective ways to implement harm reduction policies in hospital to ensure resources like needle and syringe services programs are being offered and provided reliably.

## Supporting information

**S1 File.**
(DOCX)

## Acknowledgments

Preliminary results from this project were presented at Harm Reduction International 2019 in Porto, Portugal.

PM passed away before the submission of the final version of this manuscript. TDB accepts responsibility for the integrity and validity of the data collected and analyzed.

## Author Contributions

**Conceptualization:** Thomas D. Brothers, Susan Kirkland, Patti Melanson, Lisa Barrett, Duncan Webster.

**Data curation:** Thomas D. Brothers, Kimiko Mosseler.

**Formal analysis:** Thomas D. Brothers, Kimiko Mosseler.

**Funding acquisition:** Thomas D. Brothers.

**Investigation:** Thomas D. Brothers, Kimiko Mosseler, Patti Melanson.

**Methodology:** Thomas D. Brothers, Patti Melanson, Lisa Barrett, Duncan Webster.

**Project administration:** Thomas D. Brothers.

**Resources:** Thomas D. Brothers.

**Supervision:** Thomas D. Brothers, Susan Kirkland, Lisa Barrett, Duncan Webster.

**Visualization:** Thomas D. Brothers.

**Writing – original draft:** Thomas D. Brothers.

**Writing – review & editing:** Thomas D. Brothers, Kimiko Mosseler, Susan Kirkland, Patti Melanson, Lisa Barrett, Duncan Webster.

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
