## [Decision Letter · Decision Letter 0]

16 Nov 2021

PONE-D-21-27677Unequal access to addiction care among hospitalized patients with injection drug use-associated infective endocarditisPLOS ONE

Dear Dr. Brothers,

Thank you for submitting your manuscript to PLOS ONE. After careful consideration, we feel that it has merit but does not fully meet PLOS ONE’s publication criteria as it currently stands. Therefore, we invite you to submit a revised version of the manuscript that addresses the points raised during the review process.

 In particular, both reviewers raised questions about the validity of the measurement of whether patients were offered sterile drug injecting equipment. In order for the results to be properly contextualized and their validity understood, more detail is needed on the hospital-based programs and the likelihood that you are able to capture this outcome in a valid way.

We look forward to receiving your revised manuscript.

Kind regards,

Tara Gomes

Academic Editor

PLOS ONE

Journal Requirements:

2. Please note that in order to use the direct billing option the corresponding author must be affiliated with the chosen institute. Please either amend your manuscript to change the affiliation or corresponding author, or email us at plosone@plos.org with a request to remove this option.

Reviewers' comments:

Reviewer's Responses to Questions

**Comments to the Author**

1. Is the manuscript technically sound, and do the data support the conclusions?

Reviewer #1: Partly

Reviewer #2: Yes

2. Has the statistical analysis been performed appropriately and rigorously? 

Reviewer #1: Yes

Reviewer #2: Yes

3. Have the authors made all data underlying the findings in their manuscript fully available?

Reviewer #1: No

Reviewer #2: Yes

4. Is the manuscript presented in an intelligible fashion and written in standard English?

Reviewer #1: Yes

Reviewer #2: Yes

5. Review Comments to the Author

Reviewer #1: Thank you for the opportunity to to review this manuscript, which uses a retrospective chart review design to look at several indicators of substance use care in hospital including OAT and harm reduction/sterile supply distribution. Strengths of this manuscript include some interesting findings regarding substance use in hospital as well as leaving against medical advice, and it highlights the need to improve care for people who inject drugs in hospital. My recommendations to strengthen the manuscript are as follows:

Style:

• Methods line 92 and further below, the word opioid agonist treatment is spelled out yet the acronym OAT is used above – would continue throughout

Introduction:

• I am more familiar with the term opioid agonist therapy rather than opioid agonist treatment. The former would be my suggestion.

Methods:

• Can you clarify how you searched the ICD-10 codes? Does each hospital have a database that codes discharge diagnosis?

• Overall, more detail is needed on how the chart review was performed. Are these electronic or paper records?

• The definition of untreated opioid use disorder used is “documented injection opioid use without current use of OAT”. However, opioid use disorder requires a patient to meet certain DSM-V criteria beyond just use of opioids. Anyone using opioids by injection is thereby labelled in this study as having an OUD. Would suggest clarifying the limitation that your definition is not able to differentiate between those with a DSM-V diagnosis of OUD vs. opioid use alone.

• In line 125-126, what is meant by “current use of OAT”? Can you specify the exact medications you considered to be OAT here (presumably methadone, bup/nx, and SROM based on the lines below) and on what time frame did they have to have a prescription? Was this based on a prescription database or patient self report?

• The authors mention that medical records were assessed for “documentation of hospital-based healthcare providers offering sterile drug injecting equipment”. I think the manuscript would be strengthened with further information around this. Is this something you would expect is typically documented in the chart? What healthcare providers are permitted to distribute this and is there a protocol for documentation? Were all interdisciplinary progress notes reviewed? Again, more details on the charting system and records for these hospitals would be very helpful. Given that the authors found NO documented offers of the provision of sterile injection equipment, it raises the possibility that this is simply not something that is typically documented in a chart even if a conversation is had… (especially since there is a policy for such in Saint John)

Results:

• The study is from October 2015 – March 2017. Given that this is now several years old, discussion on how things may have changed in these settings and hospitals since then (or not) would be helpful, particularly given the escalation of the overdose crisis.

• The findings in table 2 of rates of illicit or non-medical substance use in hospital and patient-initiated discharges are interesting; while a small study, this detail is a worthwhile contribution to the limited literature in this area.

Discussion:

• In line 220, the authors assert that “for patients with IDU-IE, addiction is the underlying cause of their infection”. Some patients who inject drugs may not meet criteria for an opioid use disorder (or substance use disorder) and the definition used by the authors for “OUD” may not actually meet DSM-V criteria. While I agree it is highly likely that the patients’ injection drug use caused the infection, asserting that the “addiction” is the cause is a loaded statement that cannot be readily proven in this study and would suggest rephrasing.

Reviewer #2: The authors performed a retrospective chart review of patients admitted in two Canadian Maritime hospitals with endocarditis associated with injection drug use. This study is a quality control measure looking at access to opioid agonist therapy and sterile injection equipment for inpatients. The authors demonstrate that opioid agonist treatment is not always offered to eligible patients at both hospitals, and sterile injection equipment is not offered to this cohort despite pre-existing hospital policy. The strengths of this work include availability of detailed demographics of the cohort, as well as details on opioid agonist therapy initiation in hospital, and thorough review of the available literature in the discussion. The work’s weakness is the small number of patients included.

1. Page 5, line 116-117, description of participants selection: It would be helpful to provide further description of how cases were verified. How many charts using ICD criteria alone were initially isolated? Is medical record completely electronic such that everything required to verify Duke criteria is available easily, including, for instance, blood culture results and vital signs? Or, was all the information gathered from physician documentation only, such that the authors were not able to directly verify laboratory results?

2. Page 5, line 120: The authors should explain why patients who died in hospital were excluded from the analysis.

3. Page 6, line 131: Where would the authors expect provision of sterile drug injecting equipment be documented? It may be helpful to include this detail.

4. Page 12, lines 248-254: The chief reason for difference in provision of OAT that the authors cite here is lack of or presence of a specific provider who routinely prescribes OAT. The authors should comment on other barriers to OAT for inpatients. As a physician working in the maritimes, does one need to have a special license to prescribe either suboxone or methadone? Is addiction medicine part of the standard curriculum in medical school and residency for family physicians, general internists, or anyone else? Why is it that everyone is able to treat hypertension, for instance, but a much smaller proportion of physicians are able to treat opioid addiction?

5. Page 13, lines 262-267: It would be helpful to provide further details on the inpatient policy enabling distribution of sterile injecting equipment. Is this a hospital-wide policy available on all wards? Are there specific designated staff who should be providing the equipment (RNs?)? Where is the physical equipment stored, and are there sufficient supplies? Do inpatients need to ask for the equipment to be provided, or is it supposed to be offered to all inpatients? Perhaps if this policy is available or documented in other publications it would be helpful to reference them. The authors should include their thoughts on why this policy may be ineffective, and what changes may be necessary.

6. Limitations, page 14: The authors need to address small study size and a very specific narrow focus on patients with endocarditis, whereas there are many other inpatients without endocarditis who use injection drugs and would benefit from both sterile injecting equipment and opioid agonist therapies.

7. Conclusion, page 14: The authors need to expand the conclusion and add future directions in advocacy or research. Do the authors suggest implementation of an inpatient addiction medicine service or increasing scope of the Infectious Diseases service to include formally treating addiction? Is further study is necessary to learn why the sterile injection equipment is unavailable, despite hospital policy? How can the unmet addiction care needs be met in the future?

6. PLOS authors have the option to publish the peer review history of their article (what does this mean?). If published, this will include your full peer review and any attached files.

Reviewer #1: No

Reviewer #2: No

---

## [Author Response · Author response to Decision Letter 0]

29 Nov 2021

Please see attached Word Doc file.

---

## [Editor Report · Decision Letter 1]

8 Dec 2021

PONE-D-21-27677R1Unequal access to addiction care among hospitalized patients with injection drug use-associated infective endocarditisPLOS ONE

Dear Dr. Brothers,

Thank you for submitting your manuscript to PLOS ONE. After careful consideration, we feel that it has merit but does not fully meet PLOS ONE’s publication criteria as it currently stands. Therefore, we invite you to submit a revised version of the manuscript that addresses the points raised during the review process. There is just one outstanding question that I'm hoping you can clarify. In the revised manuscript, in response to questions about recording of provision of sterile supplies in charts, the authors included the statement "However, in Saint John the needle exchange policy at the time (Fig 1) was framed as “staff protection initiative” and so it was typical to document provision of needles to patients". Is it accurate to state that it is "typical" to document provision of needles if there was no such documentation recorded in the hospital in this analysis (as per the findings of this study)? Or is this referring to documentation of provision outside of the study population in this analysis? If the authors could clarify this point, that would be helpful.

We look forward to receiving your revised manuscript.

Kind regards,

Tara Gomes

Academic Editor

PLOS ONE
---

## [Author Response · Author response to Decision Letter 1]

4 Jan 2022

Please see fully formatted response document, attached.

---

## [Editor Report · Decision Letter 2]

6 Jan 2022

PONE-D-21-27677R2Unequal access to opioid agonist treatment and sterile injecting equipment among hospitalized patients with injection drug use-associated infective endocarditisPLOS ONE

Dear Dr. Brothers,

Thank you for submitting your manuscript to PLOS ONE. After careful consideration, we feel that it has merit but does not fully meet PLOS ONE’s publication criteria as it currently stands. Therefore, we invite you to submit a revised version of the manuscript that addresses the points raised during the review process. I appreciate you taking the time to identify the new data and revise the manuscript accordingly. I just have two outstanding questions that I'm hoping you can clarify: 1. You mention that "*Of note, there was documentation from the same cardiology ward of patients being offered sterile injecting equipment and having equipment confiscated, at different times*". It isn't clear from this statement whether the confiscation occurred in the same patient who was offered the sterile equipment, or in different patients. Can you revise to clarify this point? 2. The finding of a high proportion of IE cases using hydromorphone is one that I hadn't initially noticed, and could garner some attention given the debate around hydromorphone as safer supply. From your sentence "*primarily with hydromorphone tablets and none with heroin, which is consistent with the illicit/criminalized drug supply available in the local communities*" and the accompanying citation, it seems clear that this finding is NOT due to safer supply, but due to the illicit drug supply in Nova Scotia at the time of study. Has the illicit drug supply has changed in Nova Scotia since 2017 (similar to other jurisdictions in Canada where fentanyl is now predominantly driving overdose rates)? To avoid misinterpretation of these findings (or inappropriate assignment of the hydromorphone IE risks to safer supply), I would suggest reiterating in the sentence highlighted above that this finding is consistent with the illicit drug supply in Nova Scotia *at the time of this study. *You may also want to expand this to indicate whether you believe this finding is suggestive of IE among people accessing hydromorphone for safer supply (I assume not - I don't believe that safer supply with HM would have been prevalent during your study period) to help avoid misinterpretation or misapplication of these findings to the safer supply debate. 

We look forward to receiving your revised manuscript.

Kind regards,

Tara Gomes

Academic Editor

PLOS ONE
---

## [Author Response · Author response to Decision Letter 2]

7 Jan 2022

Please see fully formatted document, attached.

---

## [Editor Report · Decision Letter 3]

13 Jan 2022

Unequal access to opioid agonist treatment and sterile injecting equipment among hospitalized patients with injection drug use-associated infective endocarditis

PONE-D-21-27677R3

Dear Dr. Brothers,

We’re pleased to inform you that your manuscript has been judged scientifically suitable for publication and will be formally accepted for publication once it meets all outstanding technical requirements.

Kind regards,

Tara Gomes

Academic Editor

PLOS ONE
---

## [Editor Report · Acceptance letter]

17 Jan 2022

PONE-D-21-27677R3 

Unequal access to opioid agonist treatment and sterile injecting equipment among hospitalized patients with injection drug use-associated infective endocarditis 

Dear Dr. Brothers:

I'm pleased to inform you that your manuscript has been deemed suitable for publication in PLOS ONE. Congratulations! Your manuscript is now with our production department. 

Kind regards, 

on behalf of

Dr. Tara Gomes 

Academic Editor

PLOS ONE